# The Effect of Different Extenders on the Quality Characteristics of European Red Deer Epididymal Sperm Stored at 5 °C

**DOI:** 10.3390/ani12192669

**Published:** 2022-10-04

**Authors:** Anna Dziekońska, Nicoletta M. Neuman, Klaudia K. Burdal, Agnieszka Wiszniewska-Łaszczych, Marek Bogdaszewski

**Affiliations:** 1Department of Animal Biochemistry and Biotechnology, University of Warmia and Mazury in Olsztyn, Oczapowskiego 5, 10-719 Olsztyn, Poland; 2Department of Veterinary Protection of Public Health, Faculty of Veterinary Medicine, University of Warmia and Mazury in Olsztyn, Oczapowskiego 14, 10-719 Olsztyn, Poland; 3Witold Stefanski Institute of Parasitology of Polish Academy of Sciences, Research Station in Kosewo, Kosewo Górne 7, 11-700 Mragowo, Poland

**Keywords:** epididymal sperm, extender, quality, apoptosis, red deer

## Abstract

**Simple Summary:**

The use of commercial extenders for storing cervid epididymal spermatozoa in a liquid state can contribute to popularizing this method of semen preservation in cervid farms. The aim of this study was to assess the effect of various extenders (Bovidyl^®^, BoviFree^®^, and BioXcell^®^) on the quality of European red deer epididymal spermatozoa stored at 5 °C. The analyses demonstrated that the tested extenders and storage time significantly affected sperm motility and functionality of different sperm structures. The Bovidyl^®^ extender containing chicken egg yolk exerted the most beneficial influence on the quality of stored sperm. Spermatozoa stored in the Bovidyl^®^ extender were characterized by satisfactory motility (above 60% on storage day 9) and the functionality over a long period of time. In turn, extenders containing plant-based ingredients and glycerol (BioXcell^®^ and BoviFree^®^) significantly decreased membrane integrity and mitochondrial activity of spermatozoa already on the first day of storage. These results indicate that European red deer epididymal spermatozoa can be stored in commercial extenders, but their viability and functionality are significantly influenced by the composition of the applied extenders. The extender containing chicken egg yolk without glycerol produced the most satisfactory results.

**Abstract:**

The aim of this study was to evaluate the effect of different extenders on the quality of European red deer epididymal sperm stored at 5 °C. Epididymal spermatozoa were collected post mortem from 10 stags and diluted with three extenders (Bovidyl^®^, BoviFree^®^, and BioXcell^®^) and stored at 5 °C. Sperm motility (TMOT), motility parameters (system CASA), plasma membrane integrity (SYBR-14^+^/PI^−^), acrosomal membrane integrity (FITC-PNA^−^/PI^−^), mitochondrial activity (JC-1/PI), viability, and apoptotic-like changes (YOPRO/PI) were evaluated. The analyses were conducted on the first and successive days of storage (D1–D7). The applied extender, storage time, and the interactions between these factors significantly (*p* < 0.001) affected most of the analyzed parameters whose values were highest in sperm samples stored in Bovidyl^®^, regardless of storage time. In Bovidyl^®^, BoviFree^®^, and BioXcell^®^ extenders, TMOT values were estimated at 83%, 63%, and 59%, respectively, on D3. The extenders significantly influenced DNA integrity on D7. The percentage of dead sperm increased from D3. The quality of stored sperm cells was significantly influenced by the extenders’ biochemical composition. BoviFree^®^ and BioXcell^®^ contain glycerol which could contribute to deteriorating the quality of spermatozoa stored at 5 °C. Sperm cells stored in the egg yolk-based extender (Bovidyl^®^) were characterized by the highest viability and functionality.

## 1. Introduction

In comparison with domestic animals, the knowledge about the reproductive biology and biotechnology of free-living animals is limited and needs to be expanded. Biotechnology methods developed for livestock are used in non-domesticated animal breeding [1,2].

Semen preservation and insemination are the most popular breeding techniques in cervid farms [3,4]. Semen preservation enables breeders to control breeding operations, preserve and apply semen from genetically superior individuals, refresh the gene pool by introducing new genetic material to a herd, and prevent inbreeding in areas where the animals colonize a small territory [4,5].

Ejaculated semen is most often preserved, but semen samples are difficult to collect from free-living animals, and epididymal sperm can be used for that purpose. Sperm harvested from the tail of epididymis differ from ejaculated spermatozoa in motility, membrane integrity, and DNA integrity [6,7], but they retain their fertilizing capacity [1,8]. Epididymal spermatozoa can be stored in both liquid and frozen state [9,10] before they are used for insemination [8,11]. Cervid spermatozoa are usually frozen in liquid nitrogen with the use of various extenders. However, this method has certain limitations because it requires specialist equipment and is more expensive than liquid storage. The fertilizing potential of cryopreserved sperm is also reduced in comparison with spermatozoa that are stored in a liquid state [12].

Cervid sperm should be preserved in a liquid state only for short-term storage [3]. The viability and functionality of Spanish red deer epididymal sperm stored in a liquid state are preserved for up to several days [9,13]. In turn, European red deer epididymal sperm can be stored in a liquid state for a long period of time [14], and their fertilizing potential is preserved for up to several days (manuscript in press). Based on these observations, the applicability of various extenders for storing the epididymal sperm of European red deer, as a representative of the cervid family, was analyzed in this study. 

Various commercial extenders are available for storing sperm in both liquid and cryopreserved state. Extender composition is one of the most important factors that influence the quality of stored spermatozoa [15]. In the literature, the effect of various extenders on the quality of red deer spermatozoa was evaluated only in cryopreserved samples [16,17].

In view of the above, this study was conducted on the assumption that commercial extenders can be effectively used to store European red deer epididymal spermatozoa in a liquid state. Sperm motility and selected sperm cell structures (membranes, mitochondria, DNA) can be evaluated to determine the applicability of commercial extenders for preserving spermatozoa in a liquid state. The results will have practical significance for cervid farms. 

Therefore, this study aimed to investigate the effect of different extenders on sperm motility, motility parameters, membrane integrity, mitochondrial activity, and the percentage of viable sperm and sperm with apoptotic-like changes in the plasma membrane in European red deer epididymal sperm stored at 5 °C.

## 2. Materials and Methods

### 2.1. Animals and Sperm Samples

Sperm samples for the study were collected post mortem from the epididymides of 10 European red deer stags (two samples were taken from each animal). The animals were older than 4 years, and their average body weight exceeded 140 kg. The stags were legally harvested during population control hunts organized in Nowe Ramuki Forest District (Region of Warmia and Mazury, Poland) during the rutting season (September–October). 

Approximately 5–8 h post mortem, the testes and the epididymides in the scrotal sacs were transported to the laboratory, where they were prepared for analyses. Spermatozoa were collected from the tail of epididymis according to a previously described procedure [14]. The tail of epididymis was incised with a scalpel, and the contents with spermatozoa were squeezed into an Eppendorf tube. The average volume of a sperm sample collected from each epididymis was 0.3 ± 0.1 mL.

The collected sperm samples were subjected to a standard evaluation. Sperm motility was assessed subjectively in each sample [18]. Only samples with motility higher than 70% were used in the study. Sperm concentration was determined in a Bürker counting chamber (Equimed-Medical Instruments, Cracow, Poland) [14]. The average sperm concentration in a sample was 5.6 × 10^9^ sperm per mL. Sperm samples obtained from the epididymides of the same individual were pooled for further analyses.

### 2.2. Procedure for Storing Spermatozoa in A Liquid State

The harvested spermatozoa were stored in a liquid state with the use of commercial extenders for preserving bovine sperm: Bovidyl^®^ (Minitub GmbH, Tiefenbach, Germany), BoviFree^®^ (Minitub GmbH, Tiefenbach, Germany), and BioXcell^®^ (IMV Technologies, L’Aigle, France). The extenders were prepared according to the manufacturers’ instructions.

Following a standard analysis of fresh sperm, sperm samples were divided into three aliquots and diluted (100 × 10^6^ spermatozoa/mL) in Bovidyl^®^, BoviFree^®^, and BioXcell^®^ extenders. The extended sperm samples were first incubated for 2 h at room temperature, and then they were transferred to a refrigerator and stored at 5 °C throughout the experiment. During the first hour of incubation at room temperature, sperm motility was assessed using the CASA system. 

### 2.3. Evaluation of Epididymal Sperm Quality Characteristics

The quality characteristics of epididymal sperm (motility and motility parameters, membrane integrity, mitochondrial activity, viability, and apoptotic-like changes in the plasma membrane) were analyzed in extended sperm samples on the first day of storage at 5 °C (after 2 h of incubation), and then every other day. All analyses were conducted until D7 for sperm extended with BoviFree^®^ and BioXcell^®^, and until D15 for sperm extended with Bovidyl^®^. 

#### 2.3.1. Assessment of Sperm Motility and Motility Parameters by the CASA System

Sperm motility and motility parameters were evaluated using the computer-assisted sperm analysis (CASA) system (HTR-IVOS 12.3; Hamilton Thorne Biosciences, Beverley, MA, USA). Before the analysis, sperm samples were diluted 1:4 with phosphate-buffered saline (PBS) to obtain 20–30 × 10^6^ spermatozoa/mL, and heated at a temperature of 37 °C for around 10 min (Thermo Block TDR-120, Göttingen, Germany). Then, aliquots of sperm samples (5 μL) were placed in a pre-warmed Makler counting chamber (Sefi-Medical Instruments Ltd., Haifa, Israel) and evaluated at 37 °C. The following parameters were analyzed: total motility (TMOT, %), progressive motility (PMOT, %), velocity average pathway (VAP, μm/s), straight line velocity (VSL, μm/s), curvilinear line velocity (VCL, μm/s), amplitude of lateral head displacement (ALH, μm), beat cross frequency (BCF, Hz), linearity coefficient (LIN, %), and straightness (STR, %). 

In the CASA analysis, sperm parameters were measured with the following settings (according to the Hamilton Thorne technical guide v. 12.3 for gazelle/deer): frame acquired—60, frame rate—60 Hz, minimum cell contrast—60, minimum cell size—5 pixels, straightness threshold—80%, low VAP cut-off—21.9 μm/s, low VSL cut-off—6.0 μm/s. A minimum of five fields per sample were assessed, with approximately 200 spermatozoa per field. 

#### 2.3.2. Plasma Membrane Integrity 

Plasma membrane integrity was assessed using SYBR-14 and PI fluorescent probes, (Live/Dead Sperm Viability Kit; Molecular Probes, Eugene, OR, USA) according to a previously described method [19] with some modifications [14]. In brief, 20 µL of sperm samples (100 × 10^6^ spermatozoa/mL) was extended in 180 μL of PBS and incubated with 2 μL of 1 mM Sybr-14 solution in HEPES-BSA solution (pH 7.4) and 2 μL of PI (2.4 μM in Tyrode’s salt solution) for 10 min at 37 °C (Thermo Block TDR-120, Göttingen, Germany). Stained sperm samples were examined under a fluorescence microscope (Olympus, model BX 41, Tokyo, Japan) at 600× magnification. Sperm with green fluorescence in the head were considered as viable with integral plasma membranes (SYBR-14^+^/PI^−^), whereas sperm with red fluorescence in the head were classified as spermatozoa with damaged membranes. At least 200 spermatozoa were counted in the quantitative assessment of plasma membrane integrity. 

#### 2.3.3. Acrosomal Membrane Integrity 

Acrosomal membrane integrity (%) was evaluated using fluorescein isothiocyanate-labeled peanut (*Arachis hypogaea*) agglutinin (FITC-PNA) staining with PI, according to a previously described method [20] with some modifications. First, sperm samples (20 µL; 100 × 10^6^ sperm/mL) were extended with 180 µL of PBS solution and incubated with 1 µL of JC-1 at 37 °C for 15 min. Then, the samples were incubated with 2 μL of FITC-PNA solution (1 mg of FITC-PNA/mL of PBS) and 2 μL of PI at 37 °C for 5 min. Subsequently, 10 μL of 10% formalin solution was added to fix the cells. Stained sperm were evaluated under a fluorescence microscope (Olympus BX 41, Tokyo, Japan) at 600× magnification. A minimum of 200 cells per slide was examined in each aliquot. The results were expressed as the percentage of sperm with acrosomal membrane integrity (not stained by FITC-PNA/PI in the sperm head; FITC-PNA^−^/PI^−^; with a visible mid-piece). 

#### 2.3.4. Mitochondrial Activity

Mitochondrial activity was assessed by examining the mitochondrial membrane potential (MMP) of sperm using a mixture of JC-1 fluorochromes with PI (Molecular Probes, Eugene, USA), according to a previously described method [14] with some modifications. Briefly, sperm samples (20 μL, 100 × 10^6^ spermatozoa/mL) were extended in 180 μL of PBS and incubated with 1 μL of JC-1 solution (1 mg JC-1/mL anhydrous dimethyl sulfoxide, DMSO) for 15 min at 37 °C. Then, sperm samples were stained with 1 μL of PI (2 μM of PI solution) for 5 min at 37 °C. A minimum of 200 spermatozoa were assessed per slide under a fluorescence microscope (Olympus BX 41, Tokyo, Japan) at 600× magnification. Sperm with an orange fluorescence in the mid-piece were considered as viable with high MMP (active mitochondria, JC-1^+^/PI^−^), whereas sperm with green fluorescence in the mid-piece and red fluorescence in the head were classified as non-viable spermatozoa with low MMP. The results were expressed as the percentage of viable sperm with high MMP. 

#### 2.3.5. DNA Integrity 

DNA integrity was analyzed according to the method proposed by Patryka et al. [21]. Samples of stored spermatozoa (50 µL each) were diluted with PBS (1:4), and 200 µL of a lysis buffer (0.1%, *v/v*; Triton X-100, 0.15M NaCl, 0.08M HCl, pH 1.4) was added to the diluted samples. The samples were left in the dark for 30 s. Then, 600 µL of acridine orange solution was added, and the samples were incubated in the dark for 3 min. For microscopic analyses, 10 µL of the sample was transferred to a slide and covered with a cover slip. Around 200 spermatozoa were examined in randomly selected fields of view under a fluorescent microscope (Olympus BX 41, Tokyo, Japan). Spermatozoa with a green head were classified as sperm with integral DNA. Spermatozoa with an orange or red head were classified as sperm with damaged DNA. The results were expressed as the percentage of sperm with integral DNA. 

#### 2.3.6. Viability and Apoptotic-like Changes in the Plasma Membrane 

The percentages of red deer spermatozoa with viable plasma membranes and apoptotic-like changes in plasma membranes were determined with the Vybrant Apoptosis Assay Kit #4 (Molecular Probes Inc., Eugene, OR, USA) according to a previously described method [22] with some modifications. First, to visualize all cells, 1 µL of JC-1 was added to 200 µL of the sperm suspension (10 × 10^6^ sperm/mL) and incubated for 10 min at 37 °C. Then, 2 μL of YO-PRO-1 solution (100 μM) and 2 μL of PI (2 μM) were added to the stained sample and incubated for 5 min at 37 °C. After incubation, stained sperm cells were examined under a fluorescence microscope (Olympus, model BX41, Tokyo, Japan) at 600× magnification. A minimum of 200 cells per slide were examined in each aliquot. Four populations of spermatozoa were identified: with an unstained head, but with a visible mid-piece (viable sperm without apoptotic-like changes; YO-PRO-1^−^/PI^−^); with a green head (sperm with apoptotic-like changes in the plasma membrane; YO-PRO-1^+^/PI^−^), with a red and green head (moribund/dying sperm cells with dual fluorescence; YO-PRO-1^+^/PI^+^), and sperm with a red head (dead/necrotic cells; YO-PRO-1^−^/PI^+^). 

### 2.4. Statistical Analysis

The data were subjected to ANOVA using the general linear model (GLM) in the Statistica software package, version 13.1 (StatSoft Incorporation, Tulsa, OK, USA). The normality of data distribution and the homogeneity of variance were analyzed using Shapiro–Wilk and Levene’s tests, respectively. The data that were not normally distributed were transformed accordingly. The percentage data were subjected to arcsine transformation, whereas sperm motility variables (VCL, VSL, VAP, ALH and BCF) were log-transformed to obtain normally distributed data before statistical analysis. All results were expressed as means ± standard error of mean (SEM). Significant main effects were compared using the Duncan’s post hoc test, and the results were regarded as significantly different at *p* < 0.05. Furthermore, the main effects of the applied extender (Bovidyl^®^, BoviFree^®^ and BioXcell^®^), day of storage (D1, D3, D5, D7), and their interactions on spermatozoa quality characteristics were analyzed using a two-way ANOVA statistical model (2 × 4). 

## 3. Results

The ANOVA revealed that the applied extender and storage time significantly affected all parameters, excluding DNA integrity and the percentage of sperm cells with apoptotic-like changes (Table 1). Storage time up to 7 days had no influence on the percentage of dying spermatozoa. In turn, the interaction between extender and storage time did not significantly affect VCL, ALH, percentage of sperm with apoptotic-like changes, or the percentage of dying spermatozoa.

### 3.1. The Effect of Different Extenders and Storage Time on Sperm Motility and Motility Parameters

The analysis of the percentage of motile sperm (TMOT) revealed that the tested extenders significantly (*p* < 0.05) influenced sperm motility from the first day of storage (D1) (Figure 1a). TMOT values were highest in spermatozoa extended with Bovidyl^®^, regardless of storage time. On the fifth day of storage (D5), TMOT values in BoviFree^®^ and BioXcell^®^ decreased below 30%. The percentage of motile spermatozoa extended with Bovidyl^®^ decreased gradually and still exceeded 50% on D15. 

The evaluated extenders also significantly affected progressive motility (PMOT) already from D1 (Figure 1b). The highest PMOT values were noted in BioXcell^®^ on D1, and in Bovidyl^®^ from D3. Beginning from D5, the analyzed parameter decreased significantly in BoviFree^®^ and BioXcell^®^ in comparison with Bovidyl^®^.

The analyzed extenders significantly affected motility parameters (VAP, VSL, and VCL) from D1 (Figure 1c–e). Motility parameters were highest in Bovidyl^®^, regardless of storage time. Beginning from D1, a significant decrease in VAP values was noted in BoviFree^®^ and BioXcell^®^ relative to Bovidyl^®^. Similar trends were noted in analyses of VSL values beginning from D5 and VCL values beginning from D7. Already on the first day of storage, the tested extenders significantly (*p* < 0.05) affected the amplitude of lateral head displacement (ALH) (Figure 1f) and beat cross frequency (BCF) (Figure 1g). The lowest ALH values were noted in BioXcell^®^, regardless of storage time. Beginning from D5, BCF values decreased significantly in BoviFree^®^ and BioXcell^®^ relative to Bovidyl^®^. On D1, the linearity coefficient (LIN) (Figure 1h) and straightness (STR) (Figure 1i) were significantly (*p* < 0.05) highest in BioXcell^®^. On successive days of storage, LIN and STR values were highest in Bovidyl^®^. On D3, LIN and STR values were lowest in BoviFree^®^.

### 3.2. The Effect of the Analyzed Extenders on Membrane Integrity, Mitochondrial Activity, and DNA Integrity

The analysis of plasma membrane integrity involving SYBR-14/PI fluorochromes (Figure 2a) revealed a gradual decrease in the percentage of viable spermatozoa with integral membranes in Bovidyl^®^, where the examined parameter still exceeded 50% on D15. In sperm stored in BoviFree^®^ and BioXcell^®^, a significant decrease in plasma membrane integrity was observed already on D1. On D5, this parameter decreased below 60% in BoviFree^®^ and below 40% in BioXcell^®^.

Similar trends were observed in the analysis of acrosomal membrane integrity involving fluorochromes FITC-PNA/PI (Figure 1b) and the mitochondrial activity analysis involving fluorochromes JC-1/PI (Figure 1c). Both parameters were highest in Bovidyl^®^, regardless of storage time. Significant differences between extenders were observed already on D1. 

In turn, significant differences in DNA integrity were not observed in spermatozoa stored in various extenders until D5 (Figure 2d). The percentage of sperm cells with integral DNA decreased significantly in BioXcell^®^ in comparison with Bovidyl^®^ and BoviFree^®^ only on D7.

### 3.3. The Effect of the Analyzed Extenders and Storage Time on Sperm Viability and Apoptotic-like Changes in the Plasma Membrane

The percentage of viable spermatozoa without apoptotic-like changes was highest in Bovidyl^®^, where the studied parameter exceeded 80% until D7 (Figure 3a). In the remaining extenders, a significant decrease (*p* < 0.05) in this parameter was observed already on D3.

An evaluation of spermatozoa with apoptotic-like changes did not reveal significant differences between extenders, regardless of storage time (Figure 3b). However, this parameter was highest in BioXcell^®^ on D5. The greatest increase in the percentage of spermatozoa with apoptotic-like changes was noted in Bovidyl^®^ on the last day of storage (D15).

The percentage of early necrotic changes in epididymal spermatozoa (Figure 3c) increased gradually on successive days of storage. On D7, the percentage of necrotic sperm cells was highest in BoviFree^®^ relative to Bovidyl^®^ and BioXcell^®^.

In turn, a significant increase in the percentage of spermatozoa with late necrotic changes (Figure 3d) was observed in BioXcell^®^ beginning from D3 and in BoviFree^®^ beginning from D7, relative to Bovidyl^®^. The percentage of dead sperm cells was highest on D7, and it was estimated at 60% in BioXcell^®^.

## 4. Discussion

The preservation of cervid spermatozoa in a liquid state can be regarded as an alternative to cryopreservation if sperm viability and fertilizing capacity can be effectively preserved over a period of several days. The process of transporting stored sperm to cervid breeding centers (which are sometimes separated by a distance of several hundred kilometers) takes several days, and preserved spermatozoa can be used in artificial insemination (AI). Epididymal spermatozoa harvested post mortem from wild animals can be particularly useful for AI because they can enrich the gene pool in cervid farms. Epididymal sperm samples collected from stags during the rutting season are characterized by high concentration, and they can be used for preparing insemination doses for AI in cervid farms. The volume and concentration of sperm in the insemination dose may vary depending on the insemination technique. In cervids, only 2–10 × 10^6^ motile spermatozoa are needed for intrauterine insemination with fresh semen, whereas the dose for vaginal insemination should contain more than 85 × 10^6^ total spermatozoa [3,4]. 

In the literature, there is a general scarcity of studies examining cervid epididymal spermatozoa stored in a liquid state with commercial extenders. The use of commercial extenders in semen preservation would substantially contribute to the popularity of this storage method in cervid farms. 

The quality characteristics of European red deer epididymal spermatozoa stored in various extenders (Bovidyl^®^, BoviFree^®^ and BioXcell^®^) were evaluated in this study. These extenders are recommended for preserving the semen of domesticated ruminants in a liquid state. BoviFree^®^ and BioXcell^®^ can also be used to cryopreserve semen. In this study, sperm motility, motility parameters, plasma membrane and acrosomal membrane integrity, mitochondrial activity, DNA integrity, the percentage of viable sperm and sperm with apoptotic-like changes were analyzed on different days of storage. These parameters can determine the suitability of gametes for assisted reproductive technologies [23].

Sperm motility and motility parameters are used to predict the fertilizing potential of spermatozoa [24,25]. The extenders tested in this study significantly affected most motility parameters already on the first day of storage, but only after 2 h of incubation at a temperature of 5 °C. No significant differences in sperm motility were observed between extenders in the first hour of incubation, probably due to the fact that the spermatozoa were affected by the extender for a short time. Sperm motility was highest in the Bovidyl^®^ extender, where TMOT values exceeded 60% until D9 or longer. These results are consistent with our previous observations [14]. In turn, most of the analyzed motility parameters decreased rapidly in BoviFree^®^ and BioXcell^®^, and on D3, TMOT values decreased below 60% and PMOT values decreased below 30% in BioXcell^®^. The observed changes in sperm motility resulted from disruptions in the function of sperm cell structures, including plasma membranes and mitochondria. Similar observations were made by other researchers [23,26]. 

In the current study, the integrity of plasma and acrosomal membranes, and mitochondrial activity were significantly influenced by the type of extender (beginning from the first day of storage), and the optimal values of these parameters were noted in Bovidyl^®^. These results could indicate that Bovidyl^®^ was most effective in preserving the functions of epididymal spermatozoa stored in a liquid state. The integrity of plasma and acrosomal membranes is an important indicator of ejaculate quality [9]. The acrosome contains enzymes that participate in the fertilization of egg cells [27]. In turn, the plasma membrane acts as a protective barrier that prevents disruptions in cell homeostasis, influences sperm motility, and determines fertilization success [28,29]. Sperm motility is also affected by the availability of energy (ATP) which is produced mainly in the mitochondria [30,31]. Disruptions in mitochondrial activity decrease mitochondrial membrane potential and compromise sperm motility [32,33], which was also observed in this study, in particular in BoviFree^®^ and BioXcell^®^ extenders. As demonstrated by Liu et al. [34], the decrease in mitochondrial activity on successive days of storage could also be attributed to growing levels of reactive oxygen species (ROS) which are generated during sperm storage. 

Spermatozoa’s fertilizing potential can be assessed based on the results of DNA integrity analyses. Chromatin damage decreases male fertility and can exert a negative effect on embryonic and fetal development [35,36]. In the present study, significant changes in DNA integrity were noted on the last days of storage only in BioXcell^®^ relative to Bovidyl^®^ and BoviFree^®^. Other researchers also demonstrated that the type of extender can significantly affect the integrity of sperm DNA [37,38]. In turn, storage time did not exert a significant influence on DNA integrity, and similar observations were made in a study of Spanish red deer epididymal spermatozoa [39] and goat spermatozoa stored in a liquid state [40]. However, some authors reported that the integrity of sperm DNA decreased significantly already after several hours of storage in a liquid state [41,42]. 

The quality of stored sperm cells can also be evaluated based on the extent of apoptotic-like changes. In the current study, the percentage of dead spermatozoa (with late-stage necrotic changes) increased significantly on successive days of storage, and similar observations were made by other authors [22,43]. Interestingly, the percentage of sperm cells with apoptotic-like changes did not differ significantly across the tested extenders, regardless of storage time. Similar observations were made in a study of boar spermatozoa stored in a liquid state [43]. In contrast, Liu et al. [34] reported a significant increase in apoptotic-like changes on successive days of storage, but the cited study involved different fluorescent probes. 

The observed differences in the qualitative characteristics of stored spermatozoa could also be attributed to variations in the composition of the tested extenders. Unlike BoviFree^®^ and BioXcell^®^, Bovidyl^®^ contains chicken’s egg yolk and does not contain glycerol. Both ingredients significantly affect sperm motility and membrane integrity. Egg yolk protects spermatozoa against cold shock, and it is widely used to preserve the semen of various animal species [44,45,46,47]. Cold shock causes considerable damage to plasma membranes, mitochondria, and acrosomes [45,47]. In other studies, egg yolk significantly improved the parameters of red deer epididymal spermatozoa after dilution and during storage in a liquid state [45,47]. However, egg yolk can lead to microbiological contamination of preserved sperm [48], which is why plant-based substitutes are increasingly used to store human and animal semen [49]. In extenders, egg yolk can be replaced by soy lecithin (a component of BioXcell^®^), lipoproteins and plant proteins (components of BoviFree^®^). However, numerous studies have shown that egg yolk-based extenders more effectively preserve sperm motility, plasma membrane and acrosomal membrane integrity, and mitochondrial membrane potential than extenders containing soy lecithin [50,51,52]. In BoviFree^®^ and BioXcell^®^, the rapid decrease in sperm motility and disruptions in plasma and acrosomal membrane integrity could also be caused by the presence of glycerol. This compound exerts a positive effect on cryopreserved spermatozoa [53,54], but it can damage cell membranes during liquid storage [55]. In spermatozoa, glycerol can damage various cell structures by inducing osmotic and non-osmotic changes [56,57,58]. These processes can also damage mitochondria that generate energy for sperm movement. The above observations could explain significant disruptions in the integrity of plasma and acrosomal membranes, and mitochondrial activity already on the first day of storage, which decreased sperm motility and accelerated their degeneration. The observed changes in the qualitative characteristics of spermatozoa stored in BoviFree^®^ and BioXcell^®^ could also point to a decrease in their fertilizing potential in comparison with sperm cells stored in Bovidyl^®^.

## 5. Conclusions

The results of this study indicate that the applied extender and storage time significantly influence the quality characteristics (excluding early apoptotic-like changes) of European red deer epididymal spermatozoa stored in a liquid state. The composition of the tested extenders most likely significantly affected sperm viability and functionality. Extenders containing glycerol and plant-based ingredients (BoviFree^®^ and BioXcell^®^) exerted negative effects on sperm quality parameters (plasma and acrosomal membrane integrity, mitochondrial activity) earlier than the Bovidyl^®^ extender which contains egg yolk, but does not contain glycerol. Bovidyl^®^ was most effective in preserving sperm cell structures and delaying the death of spermatozoa stored at 5 °C. 

## Figures and Tables

**Figure 1 animals-12-02669-f001:**
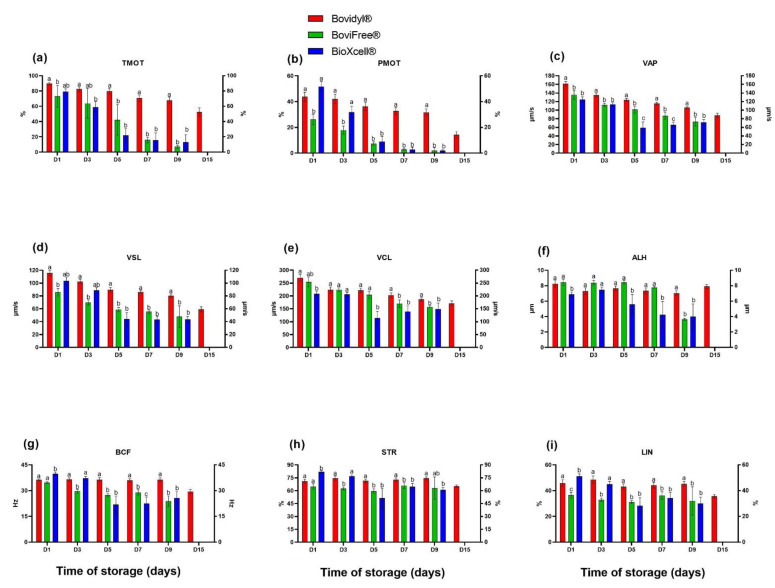
Sperm motility and motility parameters of epididymal spermatozoa stored in different extenders (Bovidyl^®^, BoviFree^®^, BioXcell^®^). (**a**) TMOT, total motility; (**b**) PMOT, progressive motility; (**c**) VAP, velocity average path; (**d**) VSL, velocity straight line; (**e**) VCL, velocity curvilinear; (**f**) ALH, amplitude of lateral head displacement (**g**) BCF, beat cross frequency (BCF), (**h**) LIN, linearity coefficient; (**i**) STR, straightness. Values represent the means of (±SEM) of extended samples of epididymal sperm collected from ten male European red deer (*n* = 10). Values with different letters (a, b, c) denote significant differences between extenders at *p* < 0.05 (Duncan’s post hoc test).

**Figure 2 animals-12-02669-f002:**
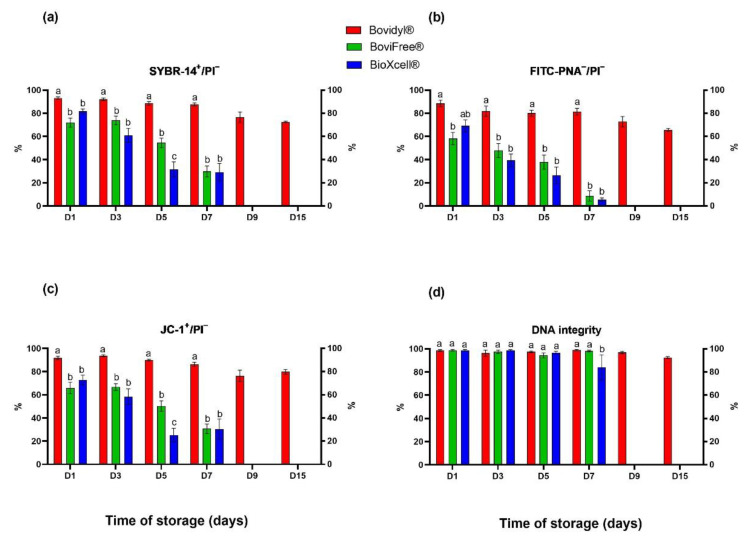
Plasma and acrosomal membrane integrity, mitochondrial membrane potential, and DNA integrity of epididymal sperm stored in different extenders (Bovidyl^®^, BoviFree^®^, BioXcell^®^). (**a**) SYBR-14^+^/PI^−^, plasma membrane integrity; (**b**) FITC-PNA^−^/PI^−^, acrosomal membrane integrity; (**c**) JC-1^+^/PI^−^, MMP, mitochondrial membrane potential; (**d**) DNA integrity. Values represent the means (± SEM) of extended samples of epididymal sperm collected from ten male European red deer (*n* = 10). Values with different letters (a, b, c) denote significant differences between extenders at *p* < 0.05 (Duncan’s post hoc test).

**Figure 3 animals-12-02669-f003:**
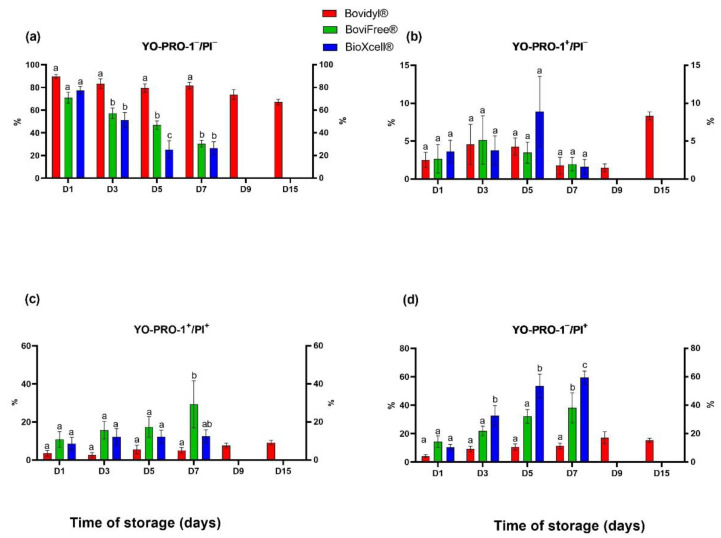
Viability and apoptotic-like changes in the plasma membrane of epididymal sperm stored in different extenders (Bovidyl^®^, BoviFree^®^, BioXcell^®^). (**a**) YO-PRO-1^−^/PI^−^, viable spermatozoa without apoptotic-like changes; (**b**) YO-PRO-1^+^/PI^−^, viable spermatozoa with apoptotic-like changes; (**c**) YO-PRO-1^+^/PI^+^ moribund spermatozoa (**d**) YO-PRO-1^−^/PI^+^, dead sperm. Values represent the means (± SEM) of extended samples of epididymal sperm collected from ten male European red deer (n = 10). Values with different letters (a, b, c) denote significant differences between extenders at *p* < 0.05 (Duncan’s post hoc test).

**Table 1 animals-12-02669-t001:** Sources of variation in the quality characteristics of stored epididymal sperm in ANOVA.

Sperm Parameters	Extender		Time of Storage		Extender × Time of Storage	
	*F*-Value	*p*-Value	*F*-Value	*p*-Value	*F*-Value	*p*-Value
TMOT	54.230	0.001	40.373	0.001	5.799	0.001
PMOT	55.854	0.001	38.605	0.001	6.544	0.001
VAP	35.840	0.001	32.735	0.001	2.381	0.034
VSL	46.555	0.001	41.351	0.001	4.133	0.001
VCL	18.774	0.001	16.350	0.001	2.093	n.s.
ALH	14.535	0.001	3.168	0.028	1.568	n.s.
BCF	13.694	0.001	13.995	0.001	6.651	0.001
LIN	15.264	0.001	7.668	0.001	3.362	0.005
STR	5.004	0.001	5.157	0.002	3.357	0.005
SYBR-14^+^/PI^−^	113.853	0.001	41.861	0.001	11.250	0.001
FITC-PNA^−^/PI^−^	114.865	0.001	29.866	0.001	6.622	0.001
DNA integrity	1.800	n.s.	2.060	n.s.	2.560	0.024
MMP	124.589	0.001	28.370	0.001	7.828	0.001
YO-PRO-1^−^/PI^−^	76.867	0.001	29.398	0.001	5.733	0.001
YO-PRO-1^+^/PI^−^	0.291	n.s.	1.386	n.s.	0.541	n.s.
YO-PRO-1^+^/PI^+^	9.892	0.001	1.406	n.s.	0.728	n.s.
YO-PRO-1^−^/PI^+^	35.130	0.001	16.344	0.001	3.633	0.001

The values are significant at *p* < 0.05; n.s., not significant; TMOT, total motility; PMOT, progressive motility; VAP, velocity average path; VSL, velocity straight line; VCL, velocity curvilinear; ALH, amplitude of lateral head displacement; BCF, beat cross frequency; LIN, linearity coefficient; STR, straightness; SYBR-14^+^/PI^−^, plasma membrane integrity; FITC-PNA^−^/PI^−^, acrosomal membrane integrity; MMP, mitochondrial membrane potential; DNA integrity; YO-PRO-1^−^/PI^−^, viable spermatozoa without apoptotic-like changes; YO-PRO-1^+^/PI^−^, spermatozoa with apoptotic-like changes; YO-PRO-1^+^/PI^+^ moribund spermatozoa; YO-PRO-1^−^/PI^+^, dead sperm.

## Data Availability

The data presented in this study are available on request from the corresponding author.

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
