# Peer review of "The Effect of Different Extenders on the Quality Characteristics of European Red Deer Epididymal Sperm Stored at 5 °C"

_animals, 2022, doi:10.3390/ani12192669_

Round 1
Reviewer 1 Report
The work is interesting and the experimental part well conducted.
Please specify the number of testis processed and the number of selected samples. What was the volume of the collected epididymal spermatozoa? The sperm concentration was measured but data were not reported. In brief: does the method of epididymal sperm collection ensure the obtainment of AI doses? This aspect shall be mentioned and discussed in the manuscript.
Please modify the formatting of the paragraph starting from line 172. Table 1 can be reduced in size by reducing the column width and the lines' interspace. Graphs can be increased in size; please check the significance letters of figure 1 graph g, h and I.
Please check the reference format of references 47, 43, 52 and 53.
Author Response
Thank you for taking the time to review the manuscript, and for all valuable remarks.

Reviewer 2 Report
I appreciate very well written and original manuscript regarding to influence of the applied extenders and storage time on qauality characteristics of European red deer epididymal spermatozoa stored in a liquid state.
All results in the study are well explained and illustrated.
I found different spacong in Chapter 2.3.5 accorfing to rest of paper, please unify this. I dont have any other reminders.
Author Response
Thank you for taking the time to review the manuscript and for the positive feedback. The spacing in subsection 2.3.5 was unified.

Reviewer 3 Report
Specific comments, critiques and suggestions for improvement are below.
Line 112: Please include the measurements of the sperm parameters for day 0. It would be beneficial to know if the different semen extenders had an immediate (i.e. within one to two hours after extending) effect on the sperm parameters.
Lines 113 to 114: What is the level or benchmark for the motility to be considered ‘significantly reduced’?
Lines 117 to 118: This reviewer understands that the samples were collected in the field and evaluated initially by a subjective analysis and then re-evaluated in the lab using a CASA system. If this is true, what was the time lag from collection to analysis with CASA?
Lines 200 to 208: Were the percentage data arcsine transformed before statistical analysis? If not, please explain.
Lines 210 to 212: This information on the repeated measures needs to be included in the statistical analysis section, not the results section.
Lines 405 to 406: This reviewer would like ‘most likely’ inserted before the word ‘significantly’ in this sentence since the authors did not specifically test the effect of composition of the semen extenders.
Author Response

(The authors gave the same response as above.)
